# Tumor-Informed Approach Improved ctDNA Detection Rate in Resected Pancreatic Cancer

**DOI:** 10.3390/ijms231911521

**Published:** 2022-09-29

**Authors:** Kazunori Watanabe, Toru Nakamura, Yasutoshi Kimura, Masayo Motoya, Shigeyuki Kojima, Tomotaka Kuraya, Takeshi Murakami, Tsukasa Kaneko, Yoshihito Shinohara, Yosuke Kitayama, Keito Fukuda, Kanako C. Hatanaka, Tomoko Mitsuhashi, Fabio Pittella-Silva, Toshikazu Yamaguchi, Satoshi Hirano, Yusuke Nakamura, Siew-Kee Low

**Affiliations:** 1Cancer Precision Medicine Center, Japanese Foundation for Cancer Research, Ariake, Koto-ku, Tokyo 135-8550, Japan; 2Department of Gastroenterological Surgery II, Faculty of Medicine, Hokkaido University, Sapporo 060-0808, Hokkaido, Japan; 3Department of Surgery, Surgical Oncology and Science, Sapporo Medical University, Sapporo 060-8556, Hokkaido, Japan; 4Department of Gastroenterology and Hepatology, Sapporo Medical University, Sapporo 060-8556, Hokkaido, Japan; 5Division of Advanced Technology & Development, BML, Inc., Kawagoe 350-1101, Saitama, Japan; 6Center for Development of Advanced Diagnostics, Hokkaido University Hospital, Sapporo 060-8648, Hokkaido, Japan; 7Department of Surgical Pathology, Hokkaido University Hospital, Sapporo 060-8648, Hokkaido, Japan; 8Laboratory of Molecular Pathology of Cancer, Faculty of Healthy Sciences and Medicine, University of Brasilia, Brasilia 70910-900, Brazil; 9National Institutes of Biomedical Innovation, Health and Nutrition, Ibaraki 567-0085, Osaka, Japan

**Keywords:** cell-free DNA, circulating tumor DNA, liquid biopsy, pancreatic cancer, tumor-informed approach, neoadjuvant therapy, next-generation sequencing, cancer prognosis

## Abstract

Pancreatic cancer is one of the cancers with very poor prognosis; there is an urgent need to identify novel biomarkers to improve its clinical outcomes. Circulating tumor DNA (ctDNA) from liquid biopsy has arisen as a promising biomarker for cancer detection and surveillance. However, it is known that the ctDNA detection rate in resected pancreatic cancer is low compared with other types of cancer. In this study, we collected paired tumor and plasma samples from 145 pancreatic cancer patients. Plasma samples were collected from 71 patients of treatment-naïve status and from 74 patients after neoadjuvant therapy (NAT). Genomic profiling of tumor DNA and plasma samples was conducted using targeted next-generation sequencing (NGS). Somatic mutations were detected in 85% (123/145) of tumors. ctDNA was detected in 39% (28/71) and 31% (23/74) of treatment-naïve and after-NAT groups, respectively, without referring to the information of tumor profiles. With a tumor-informed approach (TIA), ctDNA detection rate improved to 56% (40/71) and 36% (27/74) in treatment-naïve and after-NAT groups, respectively, with the detection rate significantly improved (*p* = 0.0165) among the treatment-naïve group compared to the after-NAT group. Cases who had detectable plasma ctDNA concordant to the corresponding tumor showed significantly shorter recurrence-free survival (RFS) (*p* = 0.0010). We demonstrated that TIA improves ctDNA detection rate in pancreatic cancer, and that ctDNA could be a potential prognostic biomarker for recurrence risk prediction

## 1. Introduction

Pancreatic cancer is the fourth leading cause of cancer death, with a 5-year survival rate of <10% in Japan, Europe, and the United States (Statistics from Japan and European populations refer to the following URLs: https://ganjoho.jp/reg_stat/statistics/stat/summary.html, https://ueg.eu/a/203, accessed on 16 May 2022.) [1]. Pancreatic cancer is often diagnosed in its advanced stages due to the lack of indicative symptoms, and approximately 80% of pancreatic cancers are surgically unresectable [2]. Furthermore, even in cases who are surgically resected, the recurrence rates within 1 and 2 years after their surgery are as high as approximately 50% and 80%, respectively [3,4]. Additionally, pancreatic cancer is expected to become the second leading cause of cancer death in the United States by 2030 [5]. Hence, the finding of biomarkers that can detect pancreatic cancer at early stages and assess recurrence risk is critically important to improve the prognosis of pancreatic cancer patients.

Mutations in *KRAS*, *TP53*, *SMAD4*, and *CDKN2A* genes are commonly found in pancreatic cancer. In particular, *KRAS* mutations are detected in approximately 90% of pancreatic cancer cases [6,7,8]. Thus, *KRAS* mutations should be important biomarkers in pancreatic cancer.

Assessment of cell-free DNA (cfDNA) through liquid biopsy using body fluids, including blood, urine, saliva, cerebrospinal fluid, and others, has been used widely in the clinical setting owing to its minimally invasive nature compared with conventional tumor tissue biopsy. cfDNA is DNA released from cells, mostly due to apoptosis or necrosis, into the blood circulation, and it is now widely known that cfDNA also contains DNA derived from tumor cells referred as circulating tumor DNA (ctDNA). The detection of ctDNA is expected to be applicable for cancer screening, detection of minimal residual disease, monitoring responses to treatment, and evaluation of drug resistance [9,10,11,12]. Monitoring of ctDNA enables its use a surrogate biomarker to assess genomic dynamic changes in tumor cells.

ctDNA detection rate increases according to disease progression [13,14,15,16], and is generally higher in advanced-stage cases than in early-stage cases. For pancreatic cancer, it was reported that the tumor volume correlated with the amount of ctDNA (Appendix A) [17,18,19,20,21,22,23,24,25]. However, compared with other cancer types, the detection rate of ctDNA in pancreatic cancer is probably lower because of the biological features of pancreatic cancer, which is characterized by a high content of extracellular matrix components, such as collagen and hyaluronan, that hinder the shedding of ctDNA into the blood circulation [26,27].

In this study, we conducted comprehensive genomic profiling of pancreatic tumor tissues by using next-generation targeted-gene sequencing (NGS). We then employed tumor-informed approach (TIA) to improve ctDNA detection rate. We also evaluated the association of ctDNA status with recurrence-free survival (RFS) of resected pancreatic cancer patients.

## 2. Results

### 2.1. Clinicopathological Characteristics of Patients in This Study

A total of 152 patients were recruited for this study. Seven patients who did not fulfill the inclusion criteria of this study were excluded from the study, as shown in Figure 1a The clinicopathological characteristics of the remaining 145 patients are summarized in Table 1 and Appendix A. Among the 145 patients, the first blood sample was collected from 71 patients before intervention (before initiation of neoadjuvant therapy (NAT): 38; upfront surgery: 33), and from 74 patients after NAT (Figure 1b). Recurrence was clinically diagnosed in 62 of the 145 patients (42.8%) during the period of evaluation (range: 1–36 months). Of the 112 patients who received NAT, 59 (52.7%) were classified as resectable, 38 (33.9%) as borderline resectable, and 15 (13.3%) as unresectable. In the upfront surgery group, all patients were classified as resectable, and the tumor extension on CT imaging was limited compared to the NAT group.

### 2.2. Genomic Profiling of Tumor Tissues and Its Association with Prognosis of Pancreatic Cancer

A total of 145 FFPE tumor tissues were subjected to genomic profiling using NGS with an Ion Ampliseq Comprehensive Cancer Panel (CCP). Sequencing parameters of NGS are summarized in Table 2 (refer Appendix A for detailed information). In brief, the median amount of genomic DNA input was 40 ng (range: 40–80 ng), and the median coverage was 236 (range: 94–618) (Appendix A). Six samples did not pass the cutoff (see Materials and Methods 4–5) for mutation detection by NGS [28]. A total of 934 mutations were detected in the tumor tissues of 139 evaluable patients, with a median of five mutations (range: 1–60) per patient (Appendix A). Figure 2a summarizes the genomic profiling of the 10 most commonly-mutated genes detected in our pancreatic cancer cases by NGS analysis. The four frequently mutated genes detected by NGS were *KRAS* (77.2%), *TP53* (52.4%), *SMAD4* (17.2%), and *CDKN2A* (8.3%). Mutation detection rates were lower in our study compared to studies from The Cancer Genome Atlas (TCGA)-PAAD study and two sets of genomic profiling reported from Asian populations (Figure 2b) [29,30,31]. One of the reasons for this low detection rate might be the inclusion of tumor tissue retrieved after NAT (Figure 2b). Among the *KRAS* mutations, p.G12D was the most common mutation (41.4%), followed by p.G12V (29.3%) and p.G12R (8.6%). Notably, p.G12R mutation was found in nearly 20% in TCGA [31], but it was approximately 10% in the three datasets from the Asian populations, including our study (Figure 2c) [29,30].

Of the 145 patients, 62 (42.8%) relapsed during a follow-up period of 1–36 months. The recurrence rate within one year after surgery was 26.9% (39/145) (range: 1–12 months). Cases with *TP53* mutations showed a significantly shorter RFS than those without *TP53* mutations detected in tumor tissue (*p* = 0.0245) (Appendix A).

### 2.3. TIA Improved ctDNA Detection Rate with Liquid Biopsy

Out of 145 cases, we received blood samples from 71 treatment-naīve cases, including 38 and 33 cases before NAT and upfront surgery, respectively (Figure 1b). For the remaining 74 cases, the first blood samples were collected after NAT (Figure 1b). All 145 plasma cell-free total nucleic acid (cfTNA) samples were sequenced using tumor-focused Oncomine Pan-Cancer Cell-Free Assay.

The sequencing parameters are summarized in Table 2. In brief, the median input of cfTNA was 20 ng (range: 4.4–20 ng), the median read coverage was 55,010 (range: 33,572–103,306), and the median molecular coverage was 4314 (range: 1200–6129) (detailed information in Appendix A). The median molecular tagging efficiency was 79% (range: 54%–100%) when we assumed 1 ng of cfDNA to be 300 haploid genomes. Clonal hematopoiesis (CH)-related mutations were excluded from ctDNA by sequencing the paired buffy coat DNA. Of note, 33 CH-related mutations were concordantly detected from plasma cfTNA and buffy DNA. CH-related mutations were most commonly detected in genes *TP53* (67%; 22 mutations from 19 patients), followed by *GNAS* (9.1%; 3 mutations from 3 patients) (Appendix A).

After the exclusion of CH-related mutations, ctDNA detection rates of the samples collected before interventions (before NAT and upfront surgery) and after NAT were 39% (28/71) and 31% (23/74), respectively. A total of 93 mutations were detected in plasma cfTNA with the threshold of 0.065% of the mutation allele frequency (MAF). ctDNA were detected most commonly in the genes *TP53*, with 46 mutations in 27 patients, and *KRAS*, with 12 mutations in 12 patients (Appendix A and Appendix A).

By using the TIA method, low numbers of copies (1–2 copies) of ctDNA or MAF below the positive threshold of 0.065% set by liquid biopsy NGS could be confirmed as tumor-derived ctDNA by referring to tumor tissue profiling. By using TIA, ctDNA detection rates improved from 39% (28/71) to 56% (40/71) in the treatment-naïve samples, and from 31% (23/74) to 36% (27/74) in patients who had received NAT (Figure 3a–c). Furthermore, the mutation detection rate guided by TIA was significantly higher in the treatment-naive samples compared to samples collected after NAT (*p* = 0.0165). ctDNA referring to TIA were most commonly detected in the genes *TP53* (60 mutations in 38 patients) and *KRAS* (22 mutations in 22 patients) (Figure 3d and Appendix A).

### 2.4. ctDNA Detected in Liquid Biopsy Based on TIA Associated with Prognosis of Pancreatic Cancer

Patients with detectable ctDNA guided by TIA were shown to have significantly shorter recurrence-free survival (RFS) than those who had no detectable ctDNA (*p* = 0.0010, Figure 4a). Specifically, patients who had detectable *KRAS* mutations from plasma cfTNA that was concordant with tumor tissue had significantly shorter RFS (*p* < 0.0001, Figure 4b) than those who were ctDNA negative. This result indicates that screening of *KRAS* ctDNA alone might help to predict patients who have a higher risk of recurrence. Notably, among the patients who had detectable mutations in both *TP53* and *KRAS* from plasma cfTNA were found to have the shortest RFS compared to those who did not have detectable *TP53* or *KRAS* mutations (*p =* 0.0005) (Figure 4c). These high-risk patients had median RFS as short as 85 days.

We also evaluated the association between clinicopathological features and ctDNA detection. Metastasis-positive cases had a significantly higher ctDNA positivity rate than metastasis-negative cases (Appendix A–e).

## 3. Discussion

Pancreatic cancer is well-known for its high mortality rate and poor prognosis, necessitating the urgent identification of clinically useful biomarkers for recurrence risk assessment. Although liquid biopsy, particularly ctDNA analysis, has been expected to be a promising and effective method for early detection of recurrence, the detection rate of ctDNA in resected pancreatic cancer patients remains low. There are several factors that cause low ctDNA detection rate in resected pancreatic cancer patients, including low tumor-cell content [20,32] as well as biological features of the tumor environment that hinder the release of ctDNA into the blood circulation [26,27].

Although we conducted molecularly barcoded ultradeep targeted sequencing for ctDNA detection with liquid biopsy to increase the likelihood of detecting very low allele frequency mutations with high confidence, the detection rate of ctDNA with NGS is relatively low (39%; 28/71) in resected pancreatic cancer patients. The detection rate is comparable with a previous study that conducted NGS analyses on 113 preoperative samples of resected pancreatic cancer, with 38% (43/113) having detectable ctDNA [21]. In this study, we exploited TIA to improve the detection rate of ctDNA from liquid biopsy. Low numbers of copies (1–2 copies) of ctDNA or MAF of ctDNA <0.065% are often classified as false positives or low-confidence calling from liquid biopsy NGS. The TIA allows the classification of these low numbers of copies of ctDNA, whether they are tumor-derived or false positives from sequencing errors caused by referring to the tumor genomic profiling. As expected, the TIA approach improved ctDNA detection rate significantly, from 39% in liquid biopsy NGS (not referring to tumor genomic profiling) to 56% using TIA among treatment-naïve samples.

As expected, a lower detection rate of ctDNA was observed among samples in the after-NAT intervention group (36% with TIA). In addition, *KRAS* and *TP53* mutations, which are involved in the carcinogenesis of pancreatic cancer, were detected at a lower rate in the samples after NAT. This result reflects the effect of NAT, and indicates the importance of plasma sample collection for prognostic evaluation before any intervention. Tumor and plasma samples after NAT could also be applied to evaluate the effectiveness of NAT for tumor-downsizing purposes, which are difficult to diagnose from imaging.

One of the unique genomic features of pancreatic cancer is that >90% of the tumors harbor *KRAS* mutations [6,7]. Importantly, since mutations on *KRAS* genes predominantly occur on codons G12/13 and Q61, we can make minimal assays to cover *KRAS* mutations by targeting these two hotspots. In fact, of the 112 cases in which *KRAS* mutations were detected in tumors, 108 had mutations at codons G12/13 and Q61. Although there are some inconsistencies in published reports [33,34,35,36,37], our study showed that patients with *KRAS* ctDNA detectable from blood plasma samples before surgery (treatment-naïve or after-NAT) had significantly shorter RFS. This result indicated screening of *KRAS* mutations with liquid biopsy before surgery might help to predict patients’ risk of recurrence. In addition to *KRAS* mutation, our results also indicated the importance of *TP53* mutations as prognostic biomarkers. Although the sample size was small, patients who carried both *KRAS* and *TP53* detected from liquid biopsy had significantly shorter RFS compared to those who had no detectable mutations from liquid biopsy (Figure 4c), with median RFS being as short as 85 days. In this context, targeted NGS screening inclusive of *KRAS* and *TP53* would be crucial. Initial screening using *KRAS* and *TP53* and incorporating the frequently mutated *SMAD4* and *CDKN2A* genes might enable their use as prognostic biomarkers for pancreatic cancer. On a clinical basis, initial screening using liquid biopsy might provide recommendations or guidelines regarding NAT for patients facing upfront surgery, which could lead to a reduction in the risk of recurrence.

The tumor-informed approach increases the sensitivity of tumor-derived mutation detection in plasma cell-free DNA. Monitoring tumor-derived mutations prospectively using liquid biopsy allows recurrence risk prediction. Signatera (commercially available ctDNA service) adopted a tumor-informed approach and developed a personalized molecular residual disease assay to assess recurrence risk. Results from Signatera predict disease recurrence in patients with various solid tumors with high accuracy [38,39,40,41].

One of the important findings from the tumor genomic analysis was that the patients with *TP53* mutations showed significantly shorter RFS than those whose tumors had no *TP53* mutation. This finding is consistent with a previous report [42]. Since *TP53* mutations are known to contribute to the progression of pancreatic cancer [6,7], the detection of *TP53* mutations in tumor tissues probably indicates the progression of the cancer cells, compared to *TP53* wild-type tumors.

The limitation of this study is that the specimens used for TIA were resected specimens, and the majority of these specimens were collected after NAT. The mutation must be detected in the tumor tissue before the application of TIA. Although we detected genomic alterations in the resected specimens after NAT, the variant allelic frequencies of 68 out of 242 mutations (28.1%) in the four major pancreatic cancer genes (*KRAS, TP53, SMAD4*, and *CDKN2A*) were less than 5%, which was lower than the common cutoff value of NGS tumor genomic profiling. The low detection rate of *KRAS* mutations in tumor tissue (77%) and the relatively low frequency of the mutated alleles might reflect the lower mutational burden in the surgically resected samples after NAT. Hence, future studies evaluating TIA with biopsy tissues before NAT intervention, refining bioinformatics pipelines in calling mutations from tumor genomic profiling with samples after NAT, or evaluating *KRAS* status using not only NGS but also droplet digital polymerase chain reaction, are warranted. Two studies reported the detection of *KRAS* mutations in >95% of the biopsy samples collected with endoscopic ultrasound-guided fine-needle aspiration [43,44]. However, these studies were conducted with a small number of cases. A study with a larger sample size should be prioritized. As pancreatic cancer is a rapidly growing cancer, identification of prognostic biomarkers at an earlier stage of the disease is crucial to improve patient outcomes.

In conclusion, we conducted a comprehensive genomic profiling of tumor tissue and liquid biopsy by utilizing targeted NGS. We highlighted the importance of adopting TIA to achieve better ctDNA detection rates from liquid biopsy NGS by referring to tumor genomic profiling analysis. Furthermore, patients with detectable *KRAS* and *TP53* ctDNA were associated with a shortened RFS, indicating ctDNA can be a potential biomarker for pancreatic cancer prognosis.

## 4. Materials and Methods

### 4.1. Patient Recruitment and Sample Collection

This study enrolled 152 patients diagnosed with pancreatic cancer and operated on at Hokkaido University Hospital and Sapporo Medical University Hospital between July 2019 and September 2021. A total of 145 patients were included in the analysis, after the exclusion of patients whose tumor tissue was unavailable, and analysis was not possible due to insufficient cfTNA concentration. The sample collection was approved by the Institutional Review Board of Hokkaido University, (IRB 18-053) Sapporo Medical University (IRB 302-240, 2019), and the Japanese Foundation for Cancer Research Review Board (IRB 2018-1016, 2020-1150). All patients provided written informed consent.

### 4.2. Clinical Sample Processing

Fourteen ml of peripheral blood from each patient was collected in EDTA-2Na tubes (Terumo). Blood plasma was centrifuged at 3000 rpm for 15 min at 4 °C within 2 h of collection. The obtained plasma sample was further centrifuged at 16,000× *g* for 10 min at 4 °C to remove cell debris. Separated plasma and buffy coats were stored at −80 °C until nucleic acid extraction.

### 4.3. cfTNA and Genomic DNA Extraction

cfTNA was extracted from 4 to 8 mL of plasma using the MagMAX Cell-Free Total Nucleic Acid Isolation kit (Thermo Fisher Scientific, Waltham, MA, USA) or NextPrep-Mag cfDNA Automated Isolation Kit (PerkinElmer, Waltham, MA, USA) according to the manufacturer’s protocol. Genomic DNA from buffy coat was extracted using the FlexiGene DNA Kit (Qiagen, Venlo, The Netherlands) or chemagic DNA Blood 400 Kit H96 (PerkinElmer, Waltham, MA, USA).

Genomic DNA of tumor tissue (both tumor and normal, if normal buffy coat DNA was absent) was extracted from ten 5 μm slices of formalin-fixed paraffin-embedded (FFPE) slides, which were macrodissected to leave only the tumor tissue, using GeneRead DNA FFPE Kit (Qiagen, Venlo, The Netherlands), according to the manufacturer’s protocol. The extracted cfTNA and genomic DNA were quantified using the Qubit DNA HS Assay Kit and Qubit DNA BR assay kit (Thermo Fisher Scientific, Waltham, MA, USA), respectively. The quality and size of extracted cfTNA were evaluated using the High Sensitivity D5000 ScreenTape (Agilent, Santa Clara, CA, USA), whereas the quality of genomic DNA was evaluated using the Genomic DNA ScreenTape (Agilent, Santa Clara, CA, USA) with TapeStation System (Agilent, Santa Clara, CA, USA).

### 4.4. Library Construction

The library for NGS evaluating ctDNA was constructed using the Oncomine Pan-Cancer Cell-Free Assay for 52 genes (Appendix A), following the manufacturer’s protocol (Thermo Fisher Scientific, Waltham, USA), using 4.4–22 ng of cfTNA and 29–31 ng of genomic DNA of buffy coat.

For tumor tissue, normal tissue, and buffy coat, libraries were prepared using 40–80 ng of extracted genomic DNA using the Ion Ampliseq Comprehensive Cancer Panel (CCP) (Thermo Fisher Scientific, Waltham, MA, USA) for 409 genes (Appendix A), including genes frequently mutated in pancreatic cancer and Ion Xpress Barcode Adapters (Thermo Fisher Scientific, Waltham, USA) following the manufacturer’s protocol (Thermo Fisher Scientific, Waltham, MA, USA). Genomic DNA from the buffy coat was fragmented to 150 bp with S220 Focused ultrasonicator (Covaris, Woburn, MA, USA) before library preparation. The quality of all constructed libraries was evaluated using the High Sensitivity D1000 ScreenTape (Agilent, Santa Clara, CA, USA).

### 4.5. Targeted NGS

The constructed libraries were subjected to template preparation using either Ion 540 Chef Kit (Thermo Fisher Scientific, Waltham, MA, USA) or Ion 550 Chef Kit (Thermo Fisher Scientific) on Ion Chef (Thermo Fisher Scientific, Waltham, MA, USA). Subsequently, sequencing was performed using the Ion GeneStudio S5 Prime System (Thermo Fisher Scientific, Waltham, MA, USA).

### 4.6. Sequencing Data Analysis

Sequence alignment and variant calling as the reference to hg19 were performed using Torrent Suite Software v5.16 and Ion Reporter v5.16 and v5.18. The workflow used for the analysis was Oncomine TagSeq Pan-Cancer Liquid Biopsy w2.5 for cfDNA and buffy coat to evaluate CH -related mutations and AmpliSeq CCP w1.2 Tumor–Normal pair for tumor genomic DNA with default parameters. The cutoff for a variant call of ctDNA was 0.065%, and the mutated allele count was equal or more than two copies. For tumor tissue alterations, samples with coverage of less than 80x were excluded. Mutations with a MAF of 5% from tumor tissue profiling were evaluated as positive after excluding single nucleotide polymorphisms detected from normal tissue. Mutations detected in a buffy coat that were concordantly detected in plasma cfTNA were evaluated as CH-related mutations. Because of the large number of NAT cases, mutations in four genes that are known to be frequently mutated in pancreatic cancer (*KRAS*, *TP53*, *SMAD4*, and *CDKN2A*) were defined as positive when detected, regardless of MAF.

### 4.7. Tumor-Informed Approach

When a mutation that was concordant with the genomic profiles of the tumor was detected in the cfDNA, it was defined as positive, even if the MAF was less than 0.065% or had only one copy of the mutated allele count.

### 4.8. Statistical Analysis

All statistical analyses were performed using JMP Pro, version 16.1.0 (SAS Institute). Pearson’s chi-square test was used for statistical analysis between clinical characteristics and the detection of ctDNA. Survival curves were constructed using the Kaplan–Meier method and evaluated with the log-rank test. Statistical significance was set at *p* < 0.05.

## Figures and Tables

**Figure 1 ijms-23-11521-f001:**
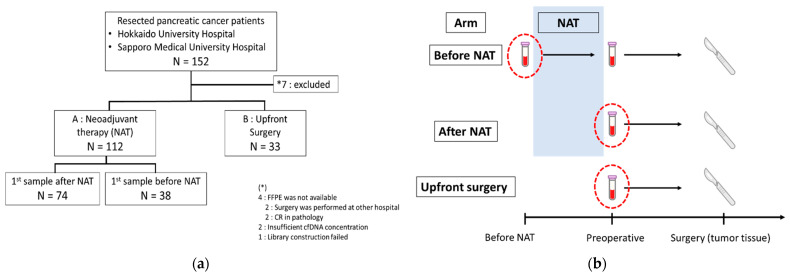
Patient enrollment and sample collection. (**a**) The consort flow diagram of patient recruitment; (**b**) sample collection timepoints of 3 different subgroups.

**Figure 2 ijms-23-11521-f002:**
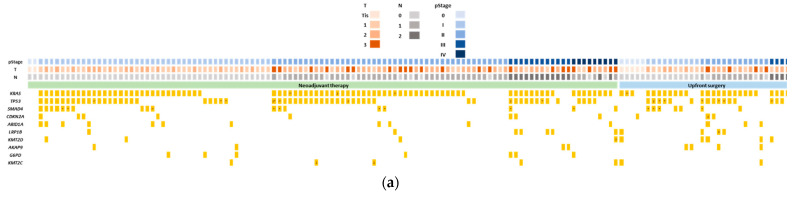
Genomic landscape of pancreatic cancer using Comprehensive Cancer Panel. (**a**) Genomic landscape of the 10 most commonly mutated genes in tumor tissue using next-generation sequencing (NGS) in pancreatic cancer. The clinical characteristics of patients are represented by the tiles at the top of the oncoplot, with details stated in the legend. *KRAS*, *TP53*, and *SMAD4* represent hotspot mutations in the liquid biopsy panel. Asterisk indicates the presence of mutations other than hotspot mutations (detected only in the tumor panel). The percentages of patients harboring *KRAS*, *TP53*, and *SMAD4* hotspot mutations were 77.2% (112/145), 44.1% (64/145), and 4.8% (7/145), respectively. All mutations below *CDKN2A* are mutations detected only in the tumor panel. (**b**) Comparison of gene mutation rates in tumor tissue against The Cancer Genome Atlas (TCGA) data and the two Asian datasets. Four mutated genes known to be commonly mutated in pancreatic cancer are shown. N indicates the number of patients. (**c**) Comparison of types of *KRAS* mutation rates in tumor tissue against TCGA data and the two Asian populations. Only the 3 most common mutations are shown. N indicates the number of mutations.

**Figure 3 ijms-23-11521-f003:**
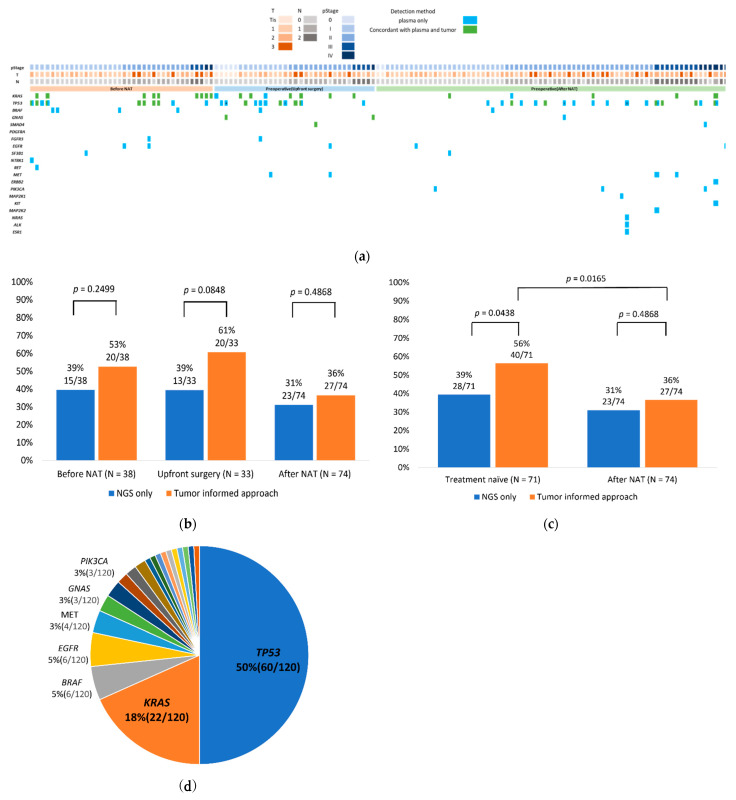
Genomic landscape of ctDNA using tumor-informed approach (TIA) from pancreatic cancer. (**a**) Mutation profiles showing all ctDNA SNVs detected from plasma cfTNA. The clinical characteristics of patients are represented by the tiles at the top of the oncoplot, with details stated in the legend. Mutations are indicated by color in the oncoplot, while numbers signify the number of SNVs for a gene per patient. SNVs that were detected from both plasma and tumor tissue are indicated in green. (**b**) ctDNA detection rates for the two approaches of the three groups. (**c**) ctDNA detection rates for the two approaches of the two groups. The detection rate significantly improved among the treatment-naïve group compared to the after-NAT group (*p* = 0.0165). *p*-values were calculated using Pearson’s chi-square test. (**d**) Overall distribution of mutations calculated as number of mutations per gene over the total 120 tumor-derived mutations, detected using TIA.

**Figure 4 ijms-23-11521-f004:**
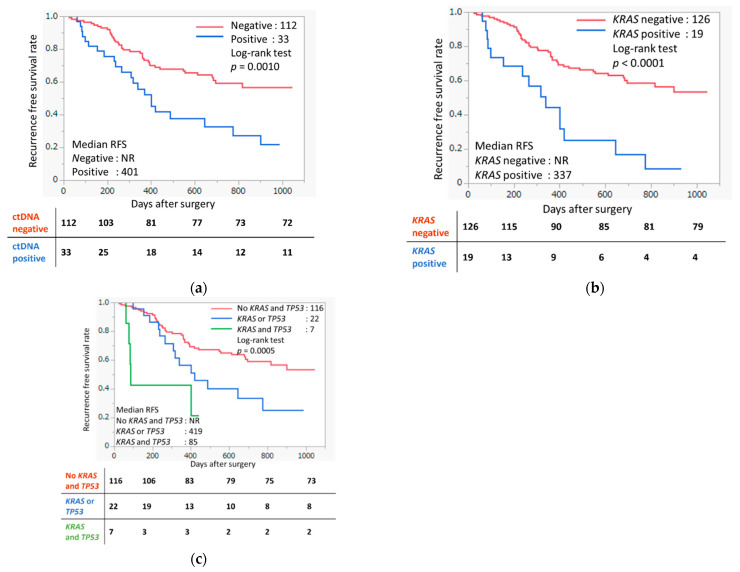
Association between the ctDNA concordant with tumor and liquid biopsy using NGS and recurrence-free survival (RFS). The differences in RFS were analyzed by log-rank test. The numbers below represent the number of recurrence-free cases at each time point (0, 200, 400, 600, 800, 1000 days). (**a**) Association between the ctDNA concordant with plasma and tumor and RFS. ctDNA positive cases had significantly shorter RFS than negative cases (*p* = 0.0010). (**b**) Association between *KRAS* detection using tumor-informed approach and RFS. Cases with detectable *KRAS* ctDNA from liquid biopsy using NGS had significantly shorter RFS (*p* < 0.0001). (**c**) Association between *KRAS* and *TP53* detection using tumor-informed approach and RFS. Cases with detectable KRAS and TP53 ctDNA from liquid biopsy using NGS had significantly shorter RFS (*p* = 0.0005).

**Table 1 ijms-23-11521-t001:** Clinical characteristics of 145 pancreatic cancer patients.

Characteristics	No. of Patients (%)
Sex	
Male	75 (51.7%)
Female	70 (48.3%)
Age	
Median (range)	71 (50–86)
Neoadjuvant therapy	
Yes	112 (77.2%)
No	33 (22.8%)
Highest preoperative CA19-9 (U/mL)	
Median (range)	80 (1–23,036.3)
Tumor location	
Head	99 (68.3%)
Body	36 (24.8%)
Tale	10 (6.9%)
Resectability	
Resectable	92 (63.4%)
Borderline resectable	38 (26.2%)
Unresectable	15 (10.3%)
UICC T stage	
Tis	7 (4.9%)
T1	48 (33.1%)
T2	64 (44.1%)
T3	26 (17.9%)
Lymph node metastasis	
Negative	73 (50.3%)
Positive	72 (49.7%)
Metastasis	
Negative	135 (93.1%)
Positive	10 (6.9%)
UICC Stage	
0	7 (4.9%)
I	54 (37.2%)
II	58 (40.0%)
III	16 (11.0%)
IV	10 (6.9%)
Recurrence	
Yes	62 (42.8%)
No	83 (57.2%)

**Table 2 ijms-23-11521-t002:** Parameters of next-generation sequencing (NGS) for 145 pair tumor tissue and liquid biopsy.

Parameters	Tumor (DNA)	Liquid Biopsy (cfTNA)
Input	40 (40–80)	20 (4.4–22.0)
Median read per sample	6,797,817 (994,453–8,115,623)	19,339,539 (12,265,807–41,115,731)
Median coverage	236 (94–618)	55,010 (33,572–103,306)
Median molecular coverage	-	4314 (1200–6129)
Mean amplicon read length (bp)	113 (83–117)	96 (83–103)
Median molecular tagging efficiency (%)	-	79% (54–100)

## Data Availability

Data will be available upon request.

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
