# Peer review of "Tumor-Informed Approach Improved ctDNA Detection Rate in Resected Pancreatic Cancer"

_ijms, 2022, doi:10.3390/ijms231911521_

Round 1

Reviewer 1 Report

The authors of the manuscript »Tumor-informed approach improved ctDNA detection rate in resected pancreatic cancer« have performed a study on prognostic significance of liquid biopsy in 145 resected pancreatic cancer patients.

In the introduction section of the manuscript the authors address clinically relevant issues of pancreatic cancer burden in modern societies, the problems of frequent tumor’s advanced stage at diagnosis as well as its poor prognosis even in case of radical surgical resection. They introduce the most common mutations found in pancreatic cancer and propose the ctDNA based liquid biopsy concept as a tool of tumor detection, surveillance as well as possible prognostic marker.

In the results section they report their findings on genetic profiling of resected pancreatic tumors and present the most common mutations they have found in tumors. The detection rate of ctDNA is reported for treatment naive and patients after neoadjuvant therapy separately. They have shown the increased ctDNA detection rate by using the tumor-informed approach in both groups of patients with stronger effect in treatment naive patients. ctDNA detection was associated with reduced recurrence free survival of patients especially in patients with detectable KRAS and TP53 mutations.

In the discussion section they nicely explain their own results and also point out some limitations of their study. However, they don't provide any comparisons of their results with the results of other similar studies which might be interesting to readers to put their findings in more objective perspective especially regarding the introduced tumor-informed approach.

The conclusions they make are nevertheless fair and straightforward.

The materials and methods section is sufficient and clear.

The used references are relevant and updated.

Reviewer 2 Report

1. In the current study, authors have enrolled a total 145 patients/samples. Out of those, 71 were before NAT intervention and 74 were after NAT. Is that all samples were from different patients or they had been collected from the same patients in a longitudinal fashion before and after NAT?  Could authors discuss if tumor informed approaches can be applied to tumor recurrence chances?  2. Line no 156, " Of the 145 patients, 62 (42.8%) relapsed during a follow-up period of 1–36 months". Do the authors identify any changes in mutation status or read counts of cfDNA in these relapsed cases?  3. Assuming KRAS mutation is identified in nearly 77% of cases. How is this tumor informed approach to detect cfDNA advantageous for remaining patients who don't have any genetic alteration in cancer cells?  4. Authors should discuss potential targetable genetic markers for initial screening of pancreatic cancer that can be tested on a clinical basis and have clinical outcome impacts.   5. Why have authors used the Oncomine Pan-Cancer Cell-Free Assay system? Does this system cover all important genetic alterations? Please provide a list of genes this system covers and if they are specific to the pancreatic cancer panel or not? 6. Materials and methods require more information.  7. Data representation/graphs are highly poor. Need improvement.  8. English grammar requires improvement. 

Reviewer 3 Report

Dear authors,

congratulations on this very well written and scientific important paper. It is a very easy to understand paper about a very important topic, which definitively adds to the general discussion about the importance of ctDNA in early tumor/progression detection. Also your recommendations on how to proceed with this type of diagnostic tool is well documented. 

There are two minor suggestions:
- out of interest: I cannot find what your NAT is, have I missed that somewhere?

- in this well written and important paper it is my opinion that the conclusion might be much more "bold" by showing/repeating the most important results, eg stating the shorter recurrence free survival correlation and mentioning the importance of KRAS/TP53 in ctDNA.  

Overal very good work in which I do not see further need for adjustments. Thank you for submitting this manuscript. 

Best wishes,

The reviewer.

Round 2

Reviewer 2 Report

Authors have addressed my comments. This article can be acceptable in IJMS